# Effects of Tebufenozide on Eggs, Larvae and Adults of *Chrysoperla carnea* (Neuroptera: Chrysopidae)

**DOI:** 10.3390/insects14060521

**Published:** 2023-06-04

**Authors:** Yurany Andrea Suarez-Lopez, Adel E. Hatem, Hani K. Aldebis, Enrique Vargas-Osuna

**Affiliations:** 1Department of Agronomy, ETSIAM, Campus Rabanales, University of Cordoba, Building C4 “Celestino Mutis”, 14071 Cordoba, Spain; 2Plant Protection Research Institute (PPRI), 7 Nadi El-Said St., Dokki, Giza 12311, Egypt

**Keywords:** green lacewing, IGRs, integrated pest management, prey preference, *Spodoptera littoralis*, sublethal effects, control agents, compatibility

## Abstract

**Simple Summary:**

Currently, alternatives for pest control in crops with lower environmental impact and risk are being sought. This study aims to evaluate the compatibility between different control alternatives/agents that can be incorporated into integrated pest management. In this case, compatible control alternatives for reducing larval populations of the harmful insect *Spodoptera littoralis* (Boisduval) (Lepidoptera: Noctuidae) were assessed. *Chrysoperla carnea* (Siemens), whose larvae are highly voracious and feed on a wide range of prey, is an attractive biological control option against a variety of insect pests. This predator resides naturally in agroecosystems or can also be added artificially by man. To find new possibilities for joint pest treatments, compatibility of insect growth regulators, IGRs (the most selective chemical insecticides), such as tebufenozide, with the predator *C. carnea* were investigated. Laboratory tests were conducted to evaluate the lethal (direct) and sublethal (indirect) effects of tebufenozide on *C. carnea*. Tebufenozide exhibited low toxicity on the developmental stages of egg, larvae and adult and is a valuable contribution to the development of integrated management programmes for *S. littoralis* that incorporate tebufenozide and *C. carnea* in a compatible and effective way, choosing the best combinations and the most appropriate time for its application.

**Abstract:**

Quantifying compatibility among control agents is essential for development of integrated pest management (IPM). *Chrysoperla carnea* (Siemens) and insect growth regulator insecticides are widely used in IPM of Lepidoptera. *C. carnea* is a generalist predator naturally present in the Mediterranean agroecosystems and bred in insectariums for commercial purposes. Here, we evaluated lethal and sublethal effects of tebufenozide on *C. carnea* under laboratory conditions. The treatment of eggs with tebufenozide 24 or 48 h after they were laid did not affect the hatching rate or survival of the neonate larvae. Toxic effects of tebufenozide on topically treated larvae was low; development times of surviving larvae and pupae decreased significantly compared with controls. In choice bioassays, a high percentage of third-instar larvae chose prey (*Spodoptera littoralis*) treated with tebufenozide in preference to untreated prey. Moreover, second-instar larvae of *C. carnea* that had previously consumed tebufenozide-treated prey (0.75 mL/L) had significantly reduced larval development time compared with controls, while longevity of surviving adults, fecundity and egg viability were unaffected. Ingestion of tebufenozide by adults of *C. carnea* at the recommended field dose had no significant effect on female fecundity, egg viability or adult longevity. Tebufenozide exhibited low toxicity towards the developmental stages of *C. carnea* and is therefore a candidate for inclusion in IPM strategies.

## 1. Introduction

Integrated pest management (IPM) is an approach based on the philosophy of combining the use of natural enemies with chemical pesticides, which are the most commonly used conventional pest control products [1,2]. The concept was later expanded to include crop resistance and cultural techniques. Integration of insecticide treatments with conservation biological control is becoming an increasingly important component of IPM [3]. A better understanding of the interactions between insecticides and the most valuable natural enemies in a crop is essential to determine whether they are compatible as pest control agents.

Global use of synthetic pesticides has increased dramatically, leading to several problems. These include toxic effects on wildlife (e.g., birds and beneficial insects) and non-target natural enemies (e.g., predators and parasites). Amarasekare and Shearer [4] determined the lethal and sublethal effects of various insecticides on two predatory lacewing species (Chrysopidae: Neuroptera). Under laboratory conditions, insecticides have the potential to reduce the efficiency of these predators. Studying only the lethal effects of pesticides does not provide the whole picture. Sublethal effects may not be sustainable for crop protection if not investigated and evaluated properly [5]. Insect growth regulators (IGRs), including tebufenozide, have been used for the control of arthropod pests in different crops. Tebufenozide is a nonsteroidal ecdysone antagonist that stimulates the moulting hormone receptor of target insect pests, especially Lepidoptera, thereby inducing premature and lethal moulting [6,7]. IGRs are used against a range of pests in agroecosystems where *C. carnea* is an important natural enemy, and ecdysone antagonists are considered the most selective and, therefore, safer [8,9,10,11]. Nevertheless, while effects on natural enemies are less likely to be lethal, sublethal effects on their life history, performance, and behavior are possible [5].

*Chrysoperla carnea* (Stephens) (Neuroptera: Chrysopidae) is an important arthropod predator of insect pests of economically important crops. It is commonly called the green lacewing, and its development to adulthood passes through the egg, larval and pupal stages. Its three larval stages are all predatory, but the last stage (L3) is the most voracious and lasts the longest [12,13]. Adults feed on pollen, plant exudates and honeydew and are essential for reproduction and maintenance of populations in the field [14]. It is considered promising as a candidate for pest management programs worldwide [15,16] due to its wide prey range and geographical distribution, resistance/tolerance to pesticides, voracious larval feeding capacity and commercial availability [15,17]. Mass (inundative) field releases of *C. carnea* have effectively controlled complex pest populations in various crops [18], increasing their widespread usefulness over other predators and specialist parasitoids [19].

The purpose of this study was to specifically determine the effects of tebufenozide on *C. carnea*: (1) the lethal effects of tebufenozide spray treatments on *C. carnea* eggs; (2) the lethal effects of tebufenozide topical treatments on larvae of *C. carnea* and the sublethal effects on development of treated survivors; (3) predator preferences when offered a choice between treated and untreated prey; (4) lethal and sublethal effects of tebufenozide on larvae of *C. carnea*, following the consumption of treated prey; and (5) effects of tebufenozide on reproduction (fecundity and egg viability) and longevity of *C. carnea* adults treated with tebufenozide via ingestion.

## 2. Materials and Methods

### 2.1. Insects and Insecticide

The *C. carnea* population was supplied by Koppert S.L. Biological Systems (Almeria, Spain), as first-stage larvae (L1). Larvae were fed with *Ephestia kuehniella* (Zeller) (Lepidoptera: Pyralidae) eggs (provided by Koppert S.L.) ad libitum and adults were fed a nutritious artificial diet made following the methods of Hassan (1975) [20].

The laboratory colony of *Spodoptera littoralis* (Boisduval) (Lepidoptera: Noctuidae) was established from larvae collected on alfalfa, *Medicago sativa* (L.) (Fabales: Fabaceae), in Cordoba province, southern Spain; the population was renewed annually with individuals taken from the same areas. Larvae were reared following the methodology of Poitout and Bues (1974) [21] and fed on an artificial diet containing alfalfa, as described by Vargas-Osuna (1985) [22].

Insect colonies and bioassays were maintained in chambers at 25 ± 2 °C, 65 ± 5% relative humidity (RH) and a photoperiod of 16:8 h light:dark (L:D) at the Laboratory of Agroforestry Entomology, University of Córdoba.

The insect growth regulator (IGR) tested was Mimic^®^ (tebufenozide 24% SC), formulated as a concentrated emulsion (Certis Europe Ltd., Alicante, Spain). It is registered in Europe for the control of lepidopteran pests of crops and forests.

### 2.2. Experiments

#### 2.2.1. Effects of Tebufenozide on *C. carnea* Eggs

Two groups of eggs (approximately 150 each) were spray treated with tebufenozide at a concentration of 0.75 mL/L. One group was 24 h old and the other group was 48 h old. The treated and control groups of eggs of each age were repeated on three occasions (*n* = 50 in total for each treatment and control) and the control group were also treated but only with distilled water and a wetting agent (tween 80 at 0.1%). The volume sprayed per group was 3 mL using a manual sprayer (manufacturer name: Cofan). After treatment, the eggs were allowed to dry for 3 h and then were placed in Petri dishes (200 mm × 20 mm) with food (*E. kuehniella* eggs) for the neonate larvae to consume as they hatched. Dishes were observed daily (at 6 h intervals) and newly hatched larvae removed. In our laboratory conditions, the eggs of *C. carnea* hatched between 4 and 5 days after oviposition. Six days after treatment those *C. carnea* eggs that had not hatched (not viable) were counted under a stereoscopic microscope. Samples of 30 larvae per replicate that hatched from 48 h old eggs and its corresponding control group were maintained individually with *E. kuehniella* eggs; 3 and 6 days after treatment larval mortality was recorded.

#### 2.2.2. Effects of Tebufenozide on *C. carnea* Larval Mortality and Development Times

Newly moulted second-instar *C. carnea* larvae (L2) were placed individually in plastic cups (30 mm in diameter and 15 mm high, with lids) and the dorsal surface of each was treated topically with 3 μL of tebufenozide in 0.1% tween 80 using a micropipette. Five concentrations were used 0.012, 0.06, 030, 0.15 and 0.75 mL/L (0.75 mL/L is the maximum recommended concentration for field use, according to Spanish authorities). Control larvae were treated using the same method but with 0.1% tween 80 in distilled water and no pesticide. There were twenty replicate larvae for each concentration and the control, and the entire experiment was repeated on three occasions (*n* = 60 in total per treatment). During each bioassay, larvae were fed ad libitum with *E. kuehniella* eggs under laboratory conditions. Larval development stage and mortality were recorded daily and from these larval development time of those surviving treatment (days between treatment and cocoon formation) and pupation period of those surviving treatment (days between cocoon formation and adult emergence) were determined. The proportion of dead larvae in each treatment was Abbott-corrected with respect to the mortality that occurred in the control [23].

#### 2.2.3. Larval *C. carnea* Preference for Untreated vs. Treated Prey

Third-instar larvae of *S. littoralis* were placed individually in transparent plastic boxes (30 mm in diameter and 15 mm high, with lids) and allowed to feed for 24 h on a leaf disk of alfalfa, *M. sativa*; the leaf disk of 5 mm diameter was either treated with 3 μL tebufenozide at a concentration of 0.75 mL/L in 0.1% aqueous tween 80 or with 0.1% aqueous tween 80 (control larvae). Only larvae that had consumed the entire leaf disk within 24 h were used as prey items. After 24 h, L3 larvae of *C. carnea* were placed individually in clean Petri dishes (60 mm) with six *S. littoralis* prey items: three *S. littoralis* larvae treated with tebufenozide and three control larvae. Treated and control *S. littoralis* larvae were distinguished from each other using paint on the thorax following the methodology of Ortiz-Moreno and Vargas-Osuna [24]. Forty replicate larvae of the predator were used, and preference and predation sequence data were gathered using continuous visual observation for 12 h.

#### 2.2.4. Effects of Consumption of Tebufenozide-Treated Prey on *C. carnea*

Second-instar larvae of *S. littoralis* were placed individually in plastic boxes (30 mm in diameter and 15 mm high, with lids) and allowed to feed for 24 h on an alfalfa leaf disk treated with 3 μL tebufenozide (in 0.1% aqueous tween 80) as described previously. Two concentrations of tebufenozide were evaluated: 0.15 mL/L and 0.75 mL/L. Control larvae were treated in the same way except that the leaf disks were treated with 3 μL of 0.1% aqueous tween 80 only. There were thirty replicate larvae per treatment and control, and the entire bioassay was repeated on three occasions. Larvae that completely consumed the alfalfa leaf disk were offered as prey to L2 larvae of *C. carnea* held individually in plastic boxes (30 mm in diameter and 15 mm high) at a rate of one prey item per larva. After 24 h, *C. carnea* larvae that had consumed the entire prey item were maintained on *E. kuehniella* eggs and observed daily for mortality and to determine: larval development time; pupation period; and the fecundity, viability and longevity of adults emerging from larvae that survived the treatment.

#### 2.2.5. Effects of Tebufenozide on the *C. carnea* Adults Treated via Ingestion

Newly emerged *C. carnea* adults were sexed, and couples (one male and one female) were confined in cylindrical transparent plastic containers (160 mm in diameter and 60 mm in height). A filter paper was placed in the upper part of each cylinder as an egg-laying substrate. A piece of plastic impregnated with diet was placed at the bottom of each cylinder, alongside a small water container (30 mm in diameter and 15 mm in height). In the treatment group, the water container was replaced with one filled with tebufenozide at a concentration of 0.75 mL/L for 24 h; the control group received only distilled water. After this period all pairs (10 per treatment and control) were transferred to clean cylinders with diet and water and evaluated every 48 h for 16 days or until the adults died. At each evaluation point, the eggs laid by each couple were removed, counted and incubated for 6 days in Petri dishes with larval food. Newly hatched larvae were counted and removed daily. After six days, the eggs that had not hatched (non-viable eggs) were counted under a stereoscopic microscope.

#### 2.2.6. Statistical Analyses

Larval and pupal development times, total fecundity and adult longevity data were analyzed using ANOVA [25] (after testing the data for normality and homoscedasticity). Comparisons of means amongst mating combinations were performed using the Tukey test, applying a 0.05 significance level in the program STATISTIX 10.0 (Analytical Software, Miller Landing, FL, USA). Data expressed as percentages (egg viability, larval mortality and prey preference) were analysed using χ^2^ test at 95%.

## 3. Results

### 3.1. Effects of Tebufenozide on C. carnea Eggs

There was no significant effect of tebufenozide treatment on viability (hatching rate) of either 24 or 48 h old eggs of *C. carnea*. In all cases, their percentage viability (% hatching) was high (Table 1). Survival of larvae hatched from 48 h treated eggs was high, and there was no mortality observed in the corresponding control.

### 3.2. Effects of Tebufenozide on C. carnea Larval Mortality and Development Time

The mortality of second-instar larvae of *C. carnea* treated topically with five doses of tebufenozide treatment was not affected, with the exception of the highest concentration (0.75 mL/L) where 13.4% of larvae died, which was significantly higher than the control (χ^2^ = 5.93, df = 1, *p* = 0.0295; Table 2). Sublethal effects of tebufenozide were studied on *C. carnea* second-instar larvae that survived topical treatment. The larval development period was significantly shorter for the treatment group (from 4.6–5.3 days) than for the control group (5.9 days) (F 5258 = 7.62, *p* < 0.0001). The pupation period was significantly shorter for the treatment group (9.2–9.6 days) than the control group (9.8 days) (F 5258 = 3.46, *p* = 0.0048). No significant differences were detected at the 0.75 mL/L concentration (Table 2).

### 3.3. Preference of C. carnea Prey Selection when Given a Choice between Tebufenozide-Treated S. littoralis Prey and Untreated Prey

Experiments on the overall prey preference and the sequence of prey selected showed that third-instar *C. carnea* larvae selected *S. littoralis* larvae treated with tebufenozide in preference to untreated *S. littoralis* larvae (χ^2^ = 7.06, df = 1, *p* = 0.0104). More *C. carnea* selected tebufenozide-treated *S. littoralis* (67.5%) as their first prey item than untreated *S. littoralis*. Significantly more *C. carnea* larvae selected tebufenozide-treated *S. littoralis* as their second prey (85%; χ^2^ = 11.17, df = 1, *p* = 0.0016) and third prey items (76.3%; χ^2^ = 5.65, df = 1, *p* = 0.0315) compared with untreated *S. littoralis* (Figure 1). Amongst the 40 replicates, there was a marked preference for consumption of tebufenozide-treated prey; 62.8% of the total prey consumed was treated and was significantly higher than the percentage of untreated prey consumed (χ^2^ = 7.06, df = 1, *p* = 0.0104).

### 3.4. Lethal and Sublethal Effects of Consuming Prey Treated with Tebufenozide on C. carnea Larvae

Mortality in *C. carnea* larvae that consumed prey treated with either of the two concentrations of tebufenozide (0.15 mL/L and 0.75 mL/L) was not significantly different to mortality in the untreated control (Table 3). Corrected mortality ranged from 4.7% to 2.1% at concentrations of 0.15 mL/L and 0.75 mL/L, respectively. The larval development time was shorter in *C. carnea* that consumed tebufenozide-treated prey than in the control group, and this effect was significant at the higher tebufenozide concentration (F2117 = 19.49, *p* = 0.0205; Table 3). In contrast, pupal duration of those surviving to emerge as adults was not significantly different amongst the treatment and control groups. For adults that survived following consumption (as larvae) of prey treated with 0.15 mL/L tebufenozide, there was a slight, but not statistically significant, reduction in fecundity and viability of eggs (Table 4); at the 0.75 mL/L concentration, surviving adults showed a slight, but not statistically significant, increase in fecundity and egg viability. In both sexes, adult longevity was not significantly affected by prior consumption of tebufenozide-treated prey as larvae.

### 3.5. Fecundity, Viability and Longevity of Adult C. carnea Treated with Tebufenozide via Ingestion

There was no mortality in adult *C. carnea* treated via ingestion with 0.75 mL/L tebufenozide; fecundity, egg viability and longevity of tebufenozide-treated adult *C. carnea* was not significantly different to the control (Table 5).

## 4. Discussion

New classes of selective pesticides that reduce risks to non-targets are increasingly replacing traditional broad-spectrum pesticides. However, it is important to understand how they can best be integrated with biological control agents to achieve sustainable management as promoted by European regulations. Beneficial organisms can be exposed to pesticides via multiple routes [26]. This study evaluated the direct and indirect effects of tebufenozide on developmental stages of the predator through exposure that was topical, via feeding on treated prey, and via ingestion. Our results also demonstrated high predator preference for consumption of prey that had been treated with tebufenozide.

Effects of insecticides on eggs and embryo survival vary according to the insect species and the active substance under study. The egg phase is usually the stage most tolerant to the action of pesticides [27]. In this study, viability of *C. carnea* eggs was not affected by tebufenozide sprays. This can be interpreted as a consequence of the protection offered to the embryo by the chorion, which is composed of impermeable sclerotized proteins that limit the entry of aqueous pesticides [27]. Egg tolerance to some insecticides has already been observed in chrysopids [28] and particularly for *C. carnea* [29]. However, the eggs of *Chrysoperla externa* (Hagen) were more susceptible to conventional insecticides (endosulfan and cypermethrin) than selective pesticides [30].

No mortality in embryos or larvae newly hatched from treated eggs was observed in our study; additionally, tebufenozide did not affect the survival of larvae emerging from 48 h old treated eggs. This is similar to results described previously for *C. externa* [31].

Our results showed that *C. carnea* second-instar larvae were slightly susceptible to topical treatment of tebufenozide. Other authors have documented similar results at maximum field-recommended concentration where tebufenozide was harmless to larvae *C. carnea* as a result of low rates of absorption and penetration in the insect integument after application (e.g., <45%, in 24 h); low penetration of tebufenozide helps to explain its nontoxicity to *C. carnea* larvae [32]. Yu [33] reported that several mechanisms may be involved in the selectivity of tebufenozide in *C. externa* larvae, and these included limited penetration through the cuticle and also alterations at the target site. Using molecular techniques, Zotti et al. [11] found in chrysopids that subtle differences in architecture of the ecdysone receptor domain may interfere with binding of tebufenozide.

Effect of insect growth regulators on non-targets can be slight compared with other chemical insecticides but this depend on active ingredient and species of insect [34,35,36,37]. For example, wexythiazox and imidacloprid were categorized as causing harmful toxicity in larvae of *C. externa* under laboratory conditions [38], while the insect growth regulator, diflubenzuron, caused significant levels of mortality in L3 larvae of *C. carnea* when topically treated [8]. Amarasekare and Shearer [4] found that the insect growth regulator, novaluron, and the pyrethroid, lambda-cyhalothrin, were both toxic to L2 larvae of *C. carnea* and *Chrysoperla johnsoni* [13].

Larval development time and pupal duration were shorter in *C. carnea* L2 larvae treated with tebufenozide than in the control. These sublethal effects could be explained by the timing of the treatment. Subsequent development of surviving larvae may be modified due to the mechanism of action of the insecticide [39,40]. However, no adverse effects were observed on larval and pupal duration and survival of *Ceraeochrysa cincta* (Neuroptera: Chrysopidae) exposed topically to tebufenozide [41].

In the choice bioassay, *C. carnea* larvae preferred *S. littoralis* prey treated with tebufenozide compared to untreated larvae. This preference may be because treated prey were weak and slow moving, making them easier to predate than the untreated larvae. It is also possible that the defensive behaviors of treated larvae were reduced, which would also make them easier to catch; this has been reported previously [26].

Using the same methodology, previous studies have shown similar results with a variety of active ingredients: preference behavior was more obvious in *C. carnea* larvae preying on *S. littoralis* treated with lufenuron (76.5% preference for treated prey), a inhibitor of chitin synthesis [29] and also significant when preying on *Xanthogaleruca luteola* (Müller) previously infected with the entomopathogenic fungus *Beauveria bassiana* (Balsamo) [42]. It is likely that the effect is not related to the active ingredient but to the intrinsic effects on physiological and growth processes of treated larvae that alter their reactions and defense behaviors. Larvae of *C. carnea* react to chemical cues produced by prey [43], and it is possible that pesticide treatment affects these cues. Navarro-Roldan and Gemeno [44] reported that the insecticide exposure altered the cuticular surface and may have induced changes in the synthesized compounds present in prey larval secretions that act defensively against generalist predators [45].

Huerta et al. [46] showed that *C. carnea* larvae preferred *S. littoralis* larvae treated with imidacloprid and natural pyrethrins compared to control larvae. In feeding assays after dimethoate treatment, seven-spot ladybirds, *Coccinella septempunctata* (L.), preferred unexposed aphid prey compared to prey treated with dimethoate [47]. Müller et al. [45] observed that worker ants, *Myrmica rubra* (Hymenoptera: Formicidae), preferred more frequently insecticide-exposed than unexposed larvae.

Preference for treated prey could have a lethal effect on *C. carnea* as seen with feeding on prey treated with lufenuron [29]. Whether feeding on insecticide-treated prey causes mortality or not, predators could also suffer a greater energy expense as a result of consuming treated prey if they have lower nutritional content compared to untreated prey.

Sublethal effects of insecticide on predators via feeding on treated prey can have an impact on the efficiency and abundance of biological control agents in the field. In this study, we evaluated sublethal effects on *C. carnea* larvae brought about by feeding on prey treated with one of two tebufenozide concentrations. Results showed that larval development time was significantly reduced in both treatments compared with the control. Suarez-Lopez et al. [29] also found significant reduction in development times of *C. carnea* larvae consuming prey treated with lufenuron (1 ml/L). Shorter larval development times as a result of exposure to insecticides decreases the overall rate of predation.

No sublethal effects were apparent for the adult *C. carnea* that survived feeding on tebufenozide-treated prey as larvae; fecundity, viability and adult longevity were not significantly different to controls. Mandour [34] found similar results for *C. carnea* larvae fed on spinosad-treated *Brevicoryne brassicae* L. aphids; there was no negative impact on mortality, fecundity and fertility of surviving adults. Giolo et al. [48] studied exposure of larvae and adults of *C. carnea* to residues of various pesticides on plant leaves; for residues of methoxyfenozide (other ecdysone agonists), they found no sublethal effects on the reproductive behaviour of adults (fecundity and fertility). Smagghe and Degheele [49] found that the predator *Podisus sagitta* (Fabricius) was able to lay eggs even after feeding on larvae of *Spodoptera exigua* (Hübner) topically treated with 20 µg/larvae of tebufenozide. In our results, the longevity of adult *C. carnea* survivors was not affected at either concentrations of tebufenozide. These results are in line with those of Amarasekare and Shearer [4] that assess the sublethal effects on larvae of two chrysopids fed on novaluron-contaminated (IGR) eggs of *E. kuehniella*; surviving adults had similar longevity to the control group.

Pesticide mortality and negative effects on behaviour of natural enemy species has been reported, mainly caused by consumption of prey contaminated with neonicotinoids. This includes the coccinellid predator, *Serangium japonicum*, and the predatory mite, *Neoseiulus californicus* (McGregor) [50], *Phytoseiulus macropilis* (Banks) [51] and the hemipteran predator, *Orius insidiosus* (Say) [52].

Several studies suggest that IGRs affect embryogenesis and reduce the egg viability of target pests but these effects differ with insect species. Beneficial arthropods exposed also respond differently depending on the method of application and the active substance used.

The safety of tebufenozide in relation to *C. carnea* adults is attributed to its low rate of absorption and penetration through the adult integument after topical application [9]. In our study, no toxicity was found when adults of *C. carnea* ingested tebufenozide-treated prey; fecundity and egg viability were not altered compared with the control. Similarly, Medina et al. [9] found that *C. carnea* adults treated topically with field rates of tebufenozide or receiving inoculum by ingestion of treated prey did not cause mortality or adverse effects on fecundity or egg hatching. To understand the lack of negative effects on reproduction parameters, Medina et al. [32] conducted a microscopic study of 8-day-old females topically treated with radioactively labelled tebufenozide and showed that oocyte growth and process of ovulation were not altered, and no radioactive substances were recovered from the ovaries and eggs.

Oral ingestion of spinosad in artificial diet rapidly killed *C. carnea* adults, and toxic symptoms included tremors, paralysis and swelling; spinosad ingestion also had a profound effect on fecundity of *C. carnea* [34]. Schneider et al. [53] found that contact-treatment of adult *Hyposoter didymator* (Thunberg) parasitoids with tebufenozide (100, 500 and 1000 mg/L) had no effect on any life parameter of their offspring and could be due to the low absorption of tebufenozide.

Neurotoxic insecticides vary in their lethal and sublethal effects on predators. Imidacloprid killed adult *C. carnea* (91.7%) within four days of ingestion without altering reproduction, and triflumuron inhibited fertility of treated adults [46]. The same results were reported by Kanna and Chandrasekaran [54], who found that longevity of *C. carnea* adults fed with emamectin at a low rate (5.0 g a.i. ha^−1^) was not significantly reduced compared to controls. Garzón et al. [35] stated that deltamethrin significantly reduced the fecundity and fertility of adult *C. carnea* when exposed to dried residues on glass surfaces.

## 5. Conclusions

In natural habitats, exposure to pesticides can be via several routes, e.g., at the time of spraying and through the food chain. This study showed that tebufenozide is not toxic to eggs of the predator *C. carnea* and of low toxicity to its larvae. Larval development times and pupal duration were both significantly reduced when treated topically. The prey preference test showed that third-instar larvae of *C. carnea* prefer treated larvae to untreated ones. Sublethal effects caused by the consumption of treated prey were mild and showed low toxicity. Reproductive parameters in adults were not affected, and there were no effects on fertility, fecundity or longevity of adults that ingested tebufenozide. It is important to consider future studies on the effects of tebufenozide on subsequent generations of the treated *Chrysoperla carnea* larvae.

Based on our results, we conclude that the use of this compound can be recommended in IPM programmes involving lacewings.

## Figures and Tables

**Figure 1 insects-14-00521-f001:**
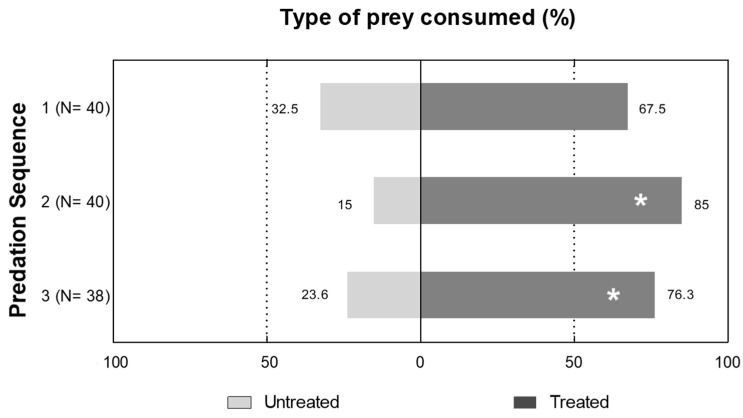
Preference of *C. carnea* larvae when given a choice between untreated and tebufenozide-treated prey. * Indicates differences between treated and untreated prey that are statistically significant at the 5 % level using the χ^2^ test. N = number of *C. carnea* larvae that fed on prey at each sequence point. All the percentage values are expressed based on the total number of *C. carnea* larvae that showed predation activity during the bioassay (n = 40).

**Table 1 insects-14-00521-t001:** Viability of 1- and 2-day-old eggs of *C. carnea* following spray treatment with tebufenozide.

Dose (mL/L)	Eggs 1 Day after Oviposition		Eggs 2 Days after Oviposition	
	*n*	% Viability ± SE	*n*	% Viability ± SE
0	130	88.58 ± 5.27	140	90.68 ± 2.16
0.75	147	88.58 ± 3.04	207	88.43 ± 2.73

*n* = Number of eggs treated. No significant differences were found between treatments. χ^2^ test.

**Table 2 insects-14-00521-t002:** Mortality of second-instar *C. carnea* larvae following topical application of tebufenozide and development time of the survivors.

Doses (mL/L)	N	Mortality		n	Larval Development Time	n	Pupal Duration
		%	Corrected ^1^		Mean ± SE (Days)		Mean ± SE (Days)
**0**	60	3.3	-	46	5.95 a ± 0.234	41	9.80 a ± 0.136
**0.012**	60	1.7	-	53	4.63 bc ± 0.175	47	9.45 bc ± 0.113
**0.06**	60	1.7	-	58	4.63 c ± 0.181	51	9.27 c ± 0.063
**0.30**	60	1.7	-	47	4.54 c ± 0.158	46	9.41 bc ± 0.073
**0.15**	60	6.7	3.4	49	5.30 b ± 0.227	43	9.51 bc ± 0.090
**0.75**	60	16.7 *	13.4	36	4.68 c ± 0.188	31	9.68 ab ± 0.149

* indicates significant differences with respect to the control at the 5% level using the χ^2^ test. ^1^ Mortality corrected using Abbott’s formula (1925). N = number of treated larvae. n = number of surviving larvae. Means followed by the same letter in a column do not differ significantly from each other at the 95% level. Test ANOVA and comparison of means-based Tukey test (*p* = 0.05).

**Table 3 insects-14-00521-t003:** Mortality of second-instar *C. carnea* larvae fed on prey treated with tebufenozide, and development time of the survivors.

Doses (mL/L)	N	Mortality		n	Larval Development Time	n	Pupal Duration
		% *	Corrected ^1^		Mean ± SE (Days)		Mean ± SE (Days)
**0**	57	7.01	-	53	8.00 a ± 0.47	40	9.58 a ± 0.24
**0.15**	59	11.86	4.78	52	6.94 ab ± 0.47	42	9.42 a ± 0.23
**0.75**	54	9.25	2.17	49	6.16 b ± 0.48	42	9.57 a ± 0.23

* No significant differences were found with respect to the control at the 5 % level using the χ^2^ test. N = number of treated larvae. n = number of surviving larvae. ^1^ Mortality corrected by Abbott’s formula (1925). Means followed by the same letter in a column do not differ significantly from each other at the 95% level. Mortality was assessed up to 8 days after treatment. Test ANOVA and comparison of means-based Tukey test.

**Table 4 insects-14-00521-t004:** The fecundity, % viability and longevity of surviving adults of *C. carnea* fed on prey treated with tebufenozide as larvae.

Doses (mL/L)	N	Fecundity (Number of Eggs)	% Viability	Female Longevity (Days)	Male Longevity (Days)
		Mean ± SE	Mean ± SE	Mean ± SE	Mean ± SE
**0**	15	112.53 ± 19.10	95.17 ± 0.88	16.53 ± 1.96	25.80 ± 1.96
**0.15**	16	92.56 ± 18.49	91.58 ± 0.88	17.56 ± 1.89	25.06 ± 1.90
**0.75**	13	142.46 ± 20.51	96.02 ± 0.95	18.69 ± 2.10	27.15 ± 2.11

N = Number of couples (replicates). No significant differences were found between treatments. Test ANOVA.

**Table 5 insects-14-00521-t005:** Fecundity and longevity of adult *C. carnea* following ingestion of tebufenozide.

Doses (mL/L)	Fecundity(Number of Eggs)		% Eggs Hatchability	Female Longevity (Days)	Male Longevity (Days)
	N	Mean ± SE	Mean ± SE	Mean ± SE	Mean ± SE
**0**	10	356.8 ± 49.75	78.1 ± 6.38	38.4 ± 7.05	40.7 ± 17.18
**0.75**	10	313.3 ± 36.94	77.5 ± 5.47	35.7 ± 10.04	43.6 ± 17.92

N = Number of couples (repetitions). No significant differences were found between treatments. Test ANOVA.

## Data Availability

Data are available upon request from the authors.

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
