# Peer review of "Effects of Tebufenozide on Eggs, Larvae and Adults of Chrysoperla carnea (Neuroptera: Chrysopidae)"

_insects, 2023, doi:10.3390/insects14060521_

Round 1

Reviewer 1 Report

The manuscript is very well. The authores have  done good laboratory work. The title chosen is appropriate and timely needed one. The objectives are reasonable. The introduction section is concise and directly emphasizes the need and the background of the study. The methodology followed in this study is standard and adaptable. The results obtained in this work have been compared with earlier works, which would be very much helpful in the understanding of principles and concepts pertaining. The section of discussion is too long and need some modification. The review of literature is extensive and exhaustive and it includes both earlier and the most recent works on this particular area. The results are therefore appreciated and warrant publication in this journal after minor revision of the manuscript

Reviewer 2 Report

The manuscript entitled “Effects of tebufenozide on eggs, larvae and adults of Chrysoperla carnea (Neuroptera: Chrysopidae) by Suarez-Lopez et al. reports results of laboratory studies on the effects of an insect growth regulator, tebufenozide on a predatory insect Chrysoperla carnea which have been used as a bio-control agent of a lepidopteran insect pest Spodoptera littoralis.

The studies provided a simple message that the IGR may be safe for the predator Chrysoperla, but further studies are required to see the effects on the subsequent generations, i.e. on the fecundity of the adults from treated Chrysoperla larvae. Faster completion of larval stages of the treated larvae or Chrysoperla larvae those were fed treated S. littoralis larvae may not have the same performance compared to the controls.

Overall the manuscript explains the information clearly but some parts of it need to be re-written, such as the simple summary.

Specific comments:

Simple summary:

Needs to be rewritten to make more sense to a general audience.

Line 12: is a lepidopteran insect..

Lines 14-16: Please make this line clearer. First, it needs to explain the function of Chrysoperla in controlling the pest insect and then mention the compatibility with IGRs

Line 16: please make it clear “first studied”. Is this a first-time study?

Lines 16-18: Laboratory tests were conducted to evaluate the lethal and sublethal effects of tebufenozide on C. carnea. Tebufenozide had a low toxicity on the developmental stages of..

Abstract:

Line 22: Determining compatibility between..?

Lines 26-28: please make it clearer

Line 36: not sure if we can tell this is a good candidate

Materials and Methods:

Line 88: Larvae were fed by Ephestia kuehniella

Line 110: manufacturer name and place of the sprayer

Line 135: are plastic boxes transparent?

Results:

Line 238-240 (and Table 4): As mentioned earlier, even in the first generation there are differences in fecundity and viability (although not statistically, but there are a big SE, which means, increasing the replications may show the statistically significant results), and it is assumed that these differences will be even more noticeable in the next generations. This emphasizes conducting further studies on the subsequent generation.

Conclusion:

A statement should be included: further studies are warranted to see the effect of tebufenozide on the subsequent generations of the treated Chrysoperla carnea larvae.

Reviewer 3 Report

The authors have evaluated the use of C. carnea as a predator for biological control of serious agricultural pest, S. littoralis. This study is a valuable contribution for integration of C. carnae in IPM programmes for Spodoptera. Additionally, the effects of chemical insecticide, tebufenozide on the developmental stages of the predator were analyzed. Data shows that S. littoralis treated with tebufenozide were preferred for consumption by C. carnea. These results have opened new possibilities for joint pest control alternatives among Lepidopteran pests.

The data looks promising and significant methodologies have been deployed which give strength to the research. The following comments would likely help the authors to improve their manuscript:

1.     Writing needs to be improved for overall text especially the abstract. Some sentences need to be rephrased to make the study more understandable. Few examples:

Lines 28-30. Eggs treated…neonate larvae.

Line 33: second instar larvae of ? It should be clear C. carnea or S. littoralis

Line 36: adults of ?

2.Keywords: IGR Integrated Management cannot be one word. It should be IGR, Integrated Pest Management or IPM

3.    Section 2.2.6, Line 179- Add reference for ANOVA: “Effect of… inherited sterility towards pest suppression”. Int J Radiat Biol. 2014 Jan;90(1):7-19. doi: 10.3109/09553002.2013.835500.

Results and Discussion section are well-written.

Author Response

  1. Writing needs to be improved for overall text especially the abstract. Some sentences need to be rephrased to make the study more understandable. We have reviewed the text to improve some sentences and we have added the recommendations.

Lines 28-30. Eggs treated…neonate larvae. We have corrected the sentence.

Line 33: second instar larvae of ? It should be clear C. carnea or S. Littoralis. We have corrected the sentence.

Line 36: adults of ? We have corrected the sentence.

  1. Keywords: IGR Integrated Management cannot be one word. It should be IGR, Integrated Pest Management or IPM. We have corrected the sentence.
  2.   Section 2.2.6, Line 179- Add reference for ANOVA: “Effect of… inherited sterility towards pest suppression”. Int J Radiat Biol. 2014 Jan;90(1):7-19. doi: 10.3109/09553002.2013.835500. We have added this reference.

Reviewer 4 Report

The study was designed adequately to quantify the toxicity on the eggs of Chrysoperla carnea following spray-treatment with tebufenozide; on larval mortality and development time of C. carnea treated topically with different doses of tebufenozide. C. carnea larvae were reported to prefer tebufenozide-treated prey over untreated prey. Furthermore, the consumption of tebufenozide-treated prey was reported not to cause significant effects on C. carnea larvae mortality, development, adult fecundity and longevity. Lastly, the study also showed that ingestion of tebufenozide did not cause a significant impact on C. carnea adult fecundity, egg hatchability and longevity. Therefore, the study concluded that tebufenozide is recommended in IPM programs involving a biological control agent, Chrysoperla carnea larvae.

The manuscript has clearly defined research questions and aims, and is generally well-written, with interesting results.  I have only some minor comments:

Materials and Methods

Row 116-117: it would be helpful if the information on how many days C. carnea eggs will hatch under normal conditions is cited here. How to make the decision if eggs are viable or not?

Row 125: Please arrange the tested doses from the highest to the lowest: 0.75, 0.30, 0.15, 0.06 and 0.012 ml/L

Results

Table 2: Please arrange data in order resulting from the lowest to the highest dose

Discussion

Row 277, 278: reference is required

Author Response

Row 116-117: it would be helpful if the information on how many days C. carnea eggs will hatch under normal conditions is cited here. We have added this information.

How to make the decision if eggs are viable or not?  The data on unhatched eggs in the control includes both the percentage of infertile eggs in the sample and the effect of the treatment methodology. Therefore, we do not believe that it is necessary to know these parameters to determine the effect of treatment with Azadirachtin.

Row 125: Please arrange the tested doses from the highest to the lowest: 0.75, 0.30, 0.15, 0.06 and 0.012 ml/L. We have made the change.

Table 2: Please arrange data in order resulting from the lowest to the highest dose. We believe that since the control data is in the first row of the tables, it is best if the concentrations are ordered from lowest to highest.

Row 277, 278: reference is required. We have add a reference.

Round 2

Reviewer 2 Report

looks good to me

Author Response

Thank you for your positive evaluation of our scientific article. We appreciate your feedback and are grateful for your recommendation to publish our work in the scientific journal.